

# Associations of physical fitness with sustained overt attention and academic performance in children with learning disabilities

Hui-Ping Chang[1], Tzu-Wen Lin[2], Yi-Hsiang Chiu[3], Chien-Chih Chou[4], Jui-Fu Chen[1] and Cheng-Chen Pan[1]

[1] Department of Physical Education and Sport Sciences, College of Sports and Recreation, National Taiwan Normal University, Taipei, Taiwan
[2] Office of Physical Education, Tamkang University, New Taipei, Taiwan
[3] Department of Physical Education, Chinese Culture University, Taipei, Taiwan
[4] Graduate Institute of Sport Pedagogy, University of Taipei, Taipei, Taiwan

## ABSTRACT

Physical fitness has consistently been linked to cognitive and academic performance, with sustained attention recognized as a key predictor of academic success (DOI 10.1123/apaq.2019-0108). However, few studies have explored whether sustained attention moderate the relationship between physical fitness and academic performance in children with learning disabilities (LD), and existing findings remain inconclusive due to certain limitations. This study investigated the moderating role of sustained attention in the association between physical fitness and academic performance among children with LD while also addressing the related limitations. This study enrolled 71 elementary school children with LD (33 girls, mean age = 11.03 years ± 0.82), who participated in the assessment of physical fitness. Additionally, sustained attention was measured using the DAUF Sustained Attention Test from the Vienna Testing System, while academic performance was assessed using Chinese language and mathematics tests'' as recommended. Bivariate analyses showed that academic performance was, as a dependent variable, significantly and positively related to physical fitness ($r = .22$ & $.24$, $p < .05$) and significantly negatively related to DAUF sustained attention ($r = -.51$ & $-.43$, $p < .01$) in children with LD. Additionally, the hierarchical regression analysis revealed that sustained attention moderated the association between physical fitness, Chinese language ($\beta = -4.03$, $p < .05$), and mathematics ($\beta = -5.00$, $p < .01$) after controlling for selected socio-demographic characteristics. These findings have major implications for child development, emphasizing the key role of physical fitness in the beneficial effects of sustained attention on academic achievement.

# INTRODUCTION

Learning disabilities (LD) are neurobiological disorders that make learning and academic work consistently difficult (*Cutting et al., 2013*; *Fletcher & Grigorenko, 2017*; *Silver et al.,*

Corresponding author
Cheng-Chen Pan,
Roger.pans@gmail.com,
cpan@ntnu.edu.tw

*2008*), rather than being caused by sensory deficits, behavioral issues, cognitive disabilities, or insufficient learning opportunities (*Bull & Scerif, 2010*). From a neurobiological perspective, students with LD have its distinctive patterns in their cognitive processing and linguistic capabilities, which create foundational differences between LD and typically developing (TD) students in their academic skill acquisition (*Cutting et al., 2013*; *Fletcher & Grigorenko, 2017*; *Parker & Boutelle, 2009*). These neural variations, mainly affecting executive functions and sustained attention, correlate strongly with reduced academic achievement (*Best, Miller & Naglieri, 2011*; *Huang et al., 2020*; *Khan & Lal, 2023*; *Peng et al., 2018*; *Silver et al., 2008*).

Modern cognitive investigations exploring cognitive mechanisms in LD have underscored the sophisticated interrelationship between executive functions and sustained attention (*Edwards et al., 2021*; *Khan & Lal, 2023*). The higher-order cognitive mechanisms rely intrinsically on sustained attention, a foundational cognitive resource that serves as a core cognitive element for prolonged mental focus, to facilitate efficient processing (*Diamond, 2013*; *Peng et al., 2018*). This interaction is particularly crucial for academic tasks such as the domains of reading comprehension and mathematical problem-solving (*Barkley, 1997*; *Rabiner, Godwin & Dodge, 2016*).

There is substantial cognitive diversity across the spectrum of LD. That reflects its shared features with other neurodevelopmental disorders. For example, each LD classification displays its own distinctive neurological patterns. Students with mathematics-specific problems may have deficits in visuospatial working memory and attentional mechanisms (*Peng et al., 2018*). Conversely, those with reading-related problems may typically show phonological processing and verbal working memory difficulties (*Gathercole et al., 2006*), as well as significant attention deficits that impact sustained focus during reading tasks (*Provazza et al., 2019*). In addition, research reveals that extensive neurobiological overlap between LD and various developmental disorders, especially ADHD, and with studies showing that 30–50% of children with ADHD meet LD criteria (*DuPaul, Gormley & Laracy, 2013*). Such observed patterns of frequent comorbidity underscore the neural mechanisms influencing attention control as well as executive functioning processes (*Faedda et al., 2019*; *Willcutt et al., 2005*). Thus, the development of effective cognitive and academic interventions should incorporate these neurological intersections.

Given this complex cognitive profile of LD, physical fitness has emerged as a potential intervention pathway for addressing cognitive and academic challenges in LD. While meta-analyses demonstrate robust associations between physical fitness and cognitive outcomes in school-aged children, children with LD face unique challenges in this domain (*Fedewa & Ahn, 2011*). They typically engage in less physical activity compared to TD peers (*Cheng et al., 2015*; *Demirci, Engin & Özmen, 2012*; *Tsai et al., 2012*), resulting in lower cardiorespiratory fitness and muscular strength (*Padmakumari & Raj, 2016*).

The relationship between physical fitness and cognitive function appears particularly important within the LD population (*Huang et al., 2020*; *Padmakumari & Raj, 2016*). Improvements in fitness may correlate with enhanced executive function and sustained attention mechanisms (*Pontifex et al., 2013*; *Reigal et al., 2020*). This shows greater benefits for children with LD than that for TD children (*Huang et al., 2020*; *Padmakumari &*

*Raj, 2016*). Some studies have reported inconsistent findings regarding the relationship between physical fitness and cognitive outcomes in children with LD. Meta-analyses have highlighted considerable variability in results (*de Greeff et al., 2018*; *Fedewa & Ahn, 2011*), with benefits varying across cognitive domains and populations. For example, *Westendorp et al. (2014)* found limited transfer effects from motor skills to academic achievement in this population. These contradictory findings contribute to the current research gap, as physical fitness has yet to be fully linked to sustained attention and subsequent academic performance in children with LD (*Muntaner-Mas et al., 2022*). Addressing this knowledge gap would enhance our understanding of cognitive profiles and attention-related deficits in children with LD.

Prior research has documented foundational knowledge about physical fitness, cognitive functions, and academic performance in LD, but substantial investigative gaps emerge. While numerous studies have established connections between these domains individually (*e.g.*, *de Greeff et al., 2018*), the integrated examination of these factors remains limited. Specifically, research examining how physical fitness might influence academic performance through cognitive pathways such as sustained attention is particularly scarce in children with LD. This gap is especially significant considering that children with LD often exhibit specific profiles of cognitive strengths and weaknesses that may uniquely influence how physical fitness impacts their academic outcomes (*Huang et al., 2020*; *Padmakumari & Raj, 2016*). Current research frequently tends to isolate these factors, either focusing on the relationship between physical fitness and cognitive functions (*Kao et al., 2017*) or examining the connections between cognition and academic performance (*Peng et al., 2018*). Indeed, research has demonstrated the benefits of physical fitness on cognitive functions in TD children (*de Greeff et al., 2018*; *Fedewa & Ahn, 2011*; *Reigal et al., 2020*) and the relationship between motor competence and academic performance in LD children (*Vuijk et al., 2011*; *Westendorp et al., 2011*). The mediating role of sustained attention across these areas remains underexplored, particularly in the neurocognitive and physical-cognitive patterns for children with LD.

To address these knowledge gaps, this study investigated whether sustained attention moderates the relationship between physical fitness and academic achievement in two fundamental academic domains (*i.e.*, Chinese language and mathematics). These academic domains were selected because of their essential academic competencies and distinct cognitive challenges in students with LD; thus, revealing the complex interplay among physical fitness and sustained attention within different types of academic tasks.

## METHODS

### Participants

All participants were recruited from schools within the same district in Taipei City, Taiwan, representing a relatively homogeneous urban educational environment. A total of 79 elementary school children with LD were recruited between 2018 and 2021 through various approaches, including posted flyers, parental referrals from local elementary schools, and orientation sessions conducted at schools to introduce the project. In Taiwan,

children are typically diagnosed with LD based on the diagnostic criteria specified in the Fourth Edition Diagnostic and Statistical Manual of Mental Disorders (*American Psychiatric Association, 1994*). These criteria, which are in accordance with Taiwan's regulations and laws pertaining to special education, include: (a) a verbal and/or performance intelligence quotient (IQ) score (above 50% of average); (b) academic achievement scores (5th to 10th percentile of the class) in Language and Mathematics; and (c) evidence of weaknesses in one or more areas in cognitive, linguistic, and/or mathematics. In this study, children with LD are identified through psycho-educational evaluations that highlight difficulties in reading, writing, and/or mathematics. They are typically two years behind their developing peers in these areas. Inclusion criteria required participants to have a formal learning disability diagnosis according to Taiwan's special education regulations. Exclusion criteria included: (1) diagnosis of autism spectrum disorder ($n = 3$ excluded) and (2) diagnosis of attention deficit hyperactivity disorder ($n = 5$ excluded). The medical history questionnaire completed by parents was used to screen for these conditions and ensure participant eligibility and safety for the physical fitness assessments. The final study consisted of 71 children with LD including 38 boys (53.52%) and 33 girls (46.48%) with a mean age of 11.03 years $\pm$ .82. The informed assent/consent for their children's participation was provided by their parents, and all procedures were in accordance with the ethical standards of the Institutional Review Board (IRB-2017-044) at the University of Taipei.

## Measures

**DAUF Sustained Attention Test:** The DAUF Sustained Attention Test is a computerized assessment designed to measure long-term attention and concentration performance through a continuous attention task. A line of five triangles were displayed on the screen, where each triangle was either pointing up or down. The triangles appeared at unpredictable moments and the participants should respond within strict time limits. The participants watched different sets of triangles while each set contained five triangles that could point up or down. They had to react quickly by pressing a button when they saw two downward facing triangles in a set. For the DAUF sustained attention performance, the mean correct response time represents the average reaction time for accurately responded stimuli, while the number of correct trials is used to calculate the accuracy rate. It is important to note that higher mean correct response time indicates poorer sustained attention (longer reaction time), while a lower number of correct trials indicates lower accuracy. Therefore, higher DAUF scores represent poorer sustained attention performance. The total duration for completing this test, including the instructional and practice phases, was approximately 20 min. Regarding the research goals, the acceptable validity of the DAUF sustained attention test was supported by previous studies (*Rostami et al., 2017*). The reliability of the DAUF sustained attention test was estimated using a Cronbach's alpha coefficient of 0.72.

  **Physical fitness**: The four subtests of the standardized test battery were used to measure physical fitness, including cardiovascular fitness (half-mile run for the fastest possible time), muscular power (standing broad jump in centimeters), muscular endurance (number of completed sit-ups in 1 min), and flexibility (sit-and-reach test). In addition to weight and

height, each participant's physical fitness was evaluated using the following procedure: the scores for each of the four fitness subsets were converted into standardized T-scores, and the overall physical fitness score was calculated as the mean of the scores across these subsets.

**Academic achievement**: Two subtests of the standardized Taiwan test battery were used to measure two aspects of the basic competence: Chinese language and mathematics. The assessment was developed by the Ministry of Education in Taiwan and is designed to evaluate fifth- and sixth-grade at elementary school students' academic competencies. The reading comprehension passages in the Chinese language test consisted of 30 multiple-choice questions and one open-ended question. These items were designed to assess the ability to acquire and read printed words to uncover facts and ideas, identify contextual relationships, understand word formation and syntax, make generalizations, and interpret information. There were 30 multiple-choice questions and one open-ended question in the mathematics test, covering topics such as number series and addressing problems related to geometry, number operations, and algebra. Standardized T-scores were applied to convert the raw scores for the academic achievement subsets, which were calculated by individual schools based on grades. The final scores were then determined as the mean of the scores from the language and mathematics tests.

**International Physical Activity Questionnaire:** The Daily Physical Activity Questionnaire (DPAQ) used two questions selected from the International Physical Activity Questionnaire-Short Form (IPAQ-SF). Overall consideration, children with LD exhibit impairments in reading and writing while having difficulty understanding the contents of the IPAQ-SF to complete each question. The study used the DPAQ where children with LD were asked three questions regarding their exercise habits such as how often they exercised (frequency), how long each activity lasted (duration), and their usual amount time of daily physical activity in a typical week. The reliability and validity of the IPAQ-SF have been previously evaluated (*Craig et al., 2003*). Overall, the IPAQ questionnaires produced repeatable data (Spearman's rho clustered around 0.8), with comparable data from short and long forms. Criterion validity had a median rho of about 0.30 (*Craig et al., 2003*).

**IQ:** The IQ test was utilized by Wechsler Intelligence Scale for Children-Third Edition (WISC-III) in this study. The participants' WISC-III scores, obtained during their fourth-grade year, were analyzed. The WISC-III, one of the most widely used and researched intelligence tests for children, consists of 13 distinct subtests, which are categorized into two scales: the Verbal Scale and the Performance Scale. According to *Wechsler (1991)*, the reliabilities of the subtests range from moderate to excellent (.61 to .92), while the consistency levels of the IQ scores and indices are reported to be very good to excellent (.80 to .97).

### Experimental procedure

The study was conducted over four days and involved five phases: (a) recruitment, (b) anthropometric measurement, (c) DAUF Sustained Attention Test, (d) assessment of physical fitness, and (e) academic achievement. During the recruitment phase, informational flyers were distributed to eight elementary schools at the onset of the
fall and sprint semesters, inviting potential participants. Those interested, along with their parents, were asked to attend an orientation session at the study location (a school). During the orientation session, the parents received a comprehensive explanation of the study's objectives, procedures, and potential benefits. Each participant and their parents subsequently signed informed consent forms, approved by our university's institutional review board. The parents also completed a detailed medical history questionnaire. On the same day, we collected anthropometric data (*i.e.,* height, weight, and body mass index (BMI)) from each participant. On the following day after anthropometric measurement, each participant completed the DAUF Sustained Attention Test, with the tests requiring a total duration of approximately 10 min. Physical fitness assessments were conducted on a separate morning on a different day, with participants instructed to refrain from strenuous physical activity on the day of assessment. Both cognitive and fitness assessments were conducted in the morning in schools, with the same experimental assistants following consistent procedures to evaluate cognitive and fitness performance. The children with LD were instructed to perform muscular fitness testing first, followed by aerobic fitness testing. Once all assessments and tests had been concluded, the participants received an NT$100 gift voucher as a token of our appreciation for their involvement in this study.

A cross-sectional design was specifically selected for this study to examine predictive relationships among multiple variables simultaneously, allowing us to investigate how physical fitness might predict academic outcomes through different pathways while accounting for the moderating role of sustained attention—relationships that are difficult to isolate in purely experimental designs.

### Statistical analysis

The moderating effect was tested for sustained attention on the relationship between physical fitness and academic performance by employing a three-step moderated hierarchical regression analysis. A general linear model is an appropriated for moderation analysis, however hierarchical regression analysis provides more useful information for examining the hypotheses. In the first step, socio-demographic characteristics (age, BMI, and physical activity) were control variables. In the second step, physical fitness and sustained attention as the independent variable was entered. In the third step, the possible interaction between physical fitness and sustained attention (the interaction term) was investigated for predicting academic performance in children with LD. A significant change in $R^2$ for the interaction term would then be taken to indicate a significant moderator effect. Since the multicollinearity and outlier problems might distort the beta terms, and they must be resolved before conducting a three-step moderated hierarchical regression analysis. Therefore, based on the theory of testing moderator effects in psychology proposed by *Frazier, Tix & Barron (2004)*, Before testing for the moderating effect, the academic performance scores of the participants in this study were standardized, which could reduce the multicollinearity problem between the interaction term and the main effect. For multicollinearity, it was detected by independent variables of correlations >0.90 and collinearity tolerance values >2.0 in this study. The standard residual statistics ranging of $\pm$ 3.3 can be detected to no outlier in this study.

**Table 1** Demographic and clinical characteristics of children with learning disabilities by LD subtype.

| Variables | Learning disability | | | | | |
| --- | --- | --- | --- | --- | --- | --- |
| | Language | | Mathematics | | | |
| | mean/n | SD/% | mean/n | SD/% | $t/\chi^2$ | $p$ |
| Age | 11.13 | 0.86 | 10.98 | 0.79 | 0.80 | 0.43 |
| LD subtype | 30 | 42.30 | 41 | 57.70 | 0.17 | 0.19 |
| Gender | | | | | | |
|   Boy | 15 | 50.00 | 23 | 56.10 | 0.26 | 0.61 |
|   Girl | 15 | 50.00 | 18 | 43.90 | | |
| BMI | 20.91 | 3.14 | 20.05 | 2.17 | 1.37 | 0.18 |
| IQ | 86.00 | 4.21 | 87.29 | 5.59 | −1.07 | 0.29 |
| Physical activity ($t$ scores) | 30.77 | 13.41 | 30.61 | 11.74 | 0.05 | 0.96 |
| Physical fitness ($t$ scores) | 50.11 | 4.46 | 50.26 | 4.70 | −0.13 | 0.89 |
|   Cardiovascular fitness | 52.14 | 9.16 | 48.41 | 10.52 | 1.56 | 0.12 |
|   Muscular power ($t$ scores) | 48.62 | 9.94 | 51.55 | 9.47 | −1.25 | 0.22 |
|   Muscular endurance ($t$ scores) | 48.87 | 9.90 | 51.34 | 9.61 | −1.06 | 0.29 |
|   Flexibility ($t$ scores) | 50.79 | 9.11 | 49.72 | 10.63 | 0.33 | 0.66 |
| DAUF sustained attention (second) | 1.00 | 0.15 | 0.94 | 0.18 | 1.55 | 0.13 |

**Notes.**
No significant differences were found between groups (all $p > 0.05$). Time-based measures: Physical activity (minutes), DAUF sustained attention (seconds). Standardized t-scores: All physical fitness measures. Learning disability types: Language impairments include dyslexia and dysgraphia.

## RESULTS

### Demographic and correlation analyses

According to Table 1, there was no significant difference among the children with learning disabilities (30 language (42.30%) and 41 mathematics (57.70%)) in terms of age, BMI, IQ, physical activity, physical fitness, and DAUF sustained attention ($t = .80$, 1.37, −1.07, .05, −.13, 1.56, −1.25, −1.06, .33, & 1.55, $p > 0.05$), as well as no significant difference in terms of types and gender of children with LD ($\chi^2 = .26$, $p > .05$).

Table 2 shows the mean and standard deviation (SD) for all variables. Zero-order correlations show that DAUF sustained attention is moderately and significantly negatively correlated with Chinese language and mathematics ($r = −.51$ & $−.43$, $p < .01$). The results revealed significant positive correlations between physical fitness and both Chinese language ($r = .22$, $p < .05$) and mathematics ($r = .24$, $p < .05$). All of the control variables, including age, physical activity, and BMI, were not significantly correlated with physical fitness, Chinese language, and mathematics. Only physical activity shows a significant negative correlation with DAUF sustained attention ($r = −.27$, $p < .05$). Then, the partial correlations among independent variables were found to range from −0.51 to 0.87, meaning they were all <0.9, and the collinearity tolerance values were found to range from −4.3 to 0.30, meaning they were all <2.0, thus indicating that no multicollinearity was detected in the analysis. Furthermore, the standard residual statistics ranged from −1.52 to 0.52, meaning they were all between +3.3 and −3.3, which indicated that no extreme outlier was found in the analysis.

**Table 2  Descriptive statistics and bivariate correlations for all study variables.**

| Variables | 1 | 2 | 3 | 4 | 5 | 6 | 7 | 8 | 9 | 10 | 11 |
|---|---|---|---|---|---|---|---|---|---|---|---|
| 1. Age | | | | | | | | | | | |
| 2. Physical activity | −0.15 | | | | | | | | | | |
| 3. BMI | −0.10 | 0.10 | | | | | | | | | |
| 4. Cardiovascular fitness (second) | 0.07 | −0.07 | −0.23$^*$ | | | | | | | | |
| 5. Muscular strength (cm) | −0.03 | −0.01 | 0.09 | −0.45$^{**}$ | | | | | | | |
| 6. Muscular endurance (times) | −0.04 | 0.17 | 0.11 | −0.31$^{**}$ | 0.30$^*$ | | | | | | |
| 7. Flexibility (cm) | −0.01 | 0.09 | −0.02 | −0.26$^*$ | 0.33$^{**}$ | 0.13 | | | | | |
| 8. Physical fitness (t scores) | −0.01 | 0.10 | −0.03 | 0.01 | 0.62$^{**}$ | 0.59$^{**}$ | 0.65$^{**}$ | | | | |
| 9. DAUF sustained attention | −0.02 | −0.27$^*$ | −0.02 | 0.34$^{**}$ | −0.11 | −0.28$^*$ | 0.00 | −0.02 | | | |
| 10. Chinese language (t scores) | −0.10 | 0.20 | 0.11 | −0.43$^{**}$ | 0.37$^{**}$ | 0.31$^{**}$ | 0.17 | 0.22$^*$ | −0.51$^{**}$ | | |
| 11. Mathematics (t scores) | −0.05 | 0.14 | 0.13 | −0.34$^{**}$ | 0.31$^{**}$ | 0.28$^*$ | 0.21 | 0.24$^*$ | −0.43$^{**}$ | 0.87$^{**}$ | |
| Mean | 11.04 | 30.68 | 20.42 | 309.70 | 128.61 | 24.54 | 23.00 | 50.19 | 0.97 | 50.08 | 49.88 |
| SD | 0.82 | 12.38 | 2.64 | 39.87 | 6.75 | 3.88 | 3.22 | 4.56 | 0.17 | 10.17 | 10.02 |

Notes.

$^*p < 0.05$.

$^{**}p < 0.01$.

Higher DAUF sustained attention scores indicate poorer performance (longer reaction times). Time-based measures: Physical activity (minutes), cardiovascular fitness and DAUF sustained attention (seconds). Distance measures: Muscular strength and flexibility (centimeters). Count measures: Muscular endurance (repetitions). Standardized t-scores: Physical fitness, Chinese language, and mathematics performance.

## Test of the moderation model of DAUF sustained attention

Table 3 indicates the moderating effects of DAUF sustained attention on the physical fitness–Chinese language. The results show that all of the control variables in the first stage did not significantly predict Chinese language ($R^2 = .05$, $p > .05$). In the second stage, both physical fitness ($\beta = .21$, $p < .05$) and DAUF sustained attention ($\beta = −.50$, $p < .01$) had significant direct effects for Chinese language with the $\Delta R^2 = .273$, $p < .01$. In the third stage, the interaction effects ($\beta = −4.03$, $p < .05$) had significant predictive for Chinese language with the $\Delta R^2 = .065$, $p < .05$, and both physical fitness and DAUF sustained attention also had significant predictive. As Fig. 1 illustrates, there is a moderating effect of DAUF sustained attention, the simple slope for low ($B = 4.84$, $p < .01$) and high ($B = .49$, $p < .05$) DAUF sustained attention are significant. The results indicate that DAUF sustained attention has a moderating effect. Children with LD who have better physical fitness tend to achieve higher Chinese language scores if their DAUF sustained attention levels are low.

Table 4 indicates the moderating effects of DAUF sustained attention on the physical fitness–mathematics. Results show that the control variables in the first stage had no significant predictive for mathematics ($R^2 = .033$, $p > .05$). In the second stage, both physical fitness ($\beta = .12$, $p < .05$) and DAUF sustained attention ($\beta = −.43$, $p < .01$) had significant direct effects for mathematics with the $\Delta R^2 = .233$, $p < .01$. In the third stage, the interaction effects ($\beta = −5.00$, $p < .01$) had significant predictive for mathematics with the $\Delta R^2 = .100$, $p < .05$, and both physical fitness and DAUF sustained attention also had significant predictive. As Fig. 2 illustrates, there is a moderating effect of DAUF sustained attention, the simple slope for low ($B = 5.85$, $p < .01$) and high ($B = .54$, $p < .05$) DAUF sustained attention are significant. The results indicate that DAUF sustained attention has

**Table 3  Summary results of the moderating effects for Chinese language.**

| Variables | Step 1: control variable | | | Step 2: direct effect | | | | Step 3: interaction effect | | | |
|---|---|---|---|---|---|---|---|---|---|---|---|
| | $\beta$ | $t$-value | $R^2$ | $\beta$ | $t$-value | $R^2$ | $\Delta R^2$ | $\beta$ | $t$-value | $R^2$ | $\Delta R^2$ |
| Age | −0.07 | −0.54 | | −0.10 | −0.93 | | | −0.09 | −0.92 | | |
| BMI | 0.08 | 0.68 | | 0.09 | 0.91 | | | 0.13 | 1.30 | | |
| Physical activity | 0.18 | 1.49 | 0.051 | 0.02 | 0.15 | 0.324 | 0.273** | −0.01 | −0.08 | 0.623 | 0.065* |
| Physical fitness | | | | 0.21 | 2.04* | | | 2.17 | 2.86** | | |
| DAUF sustained attention | | | | −0.50 | −4.71** | | | 3.05 | 2.23* | | |
| Interaction | | | | | | | | −4.03 | −2.61* | | |

Notes.
*$p < 0.05$.
**$p < 0.01$.

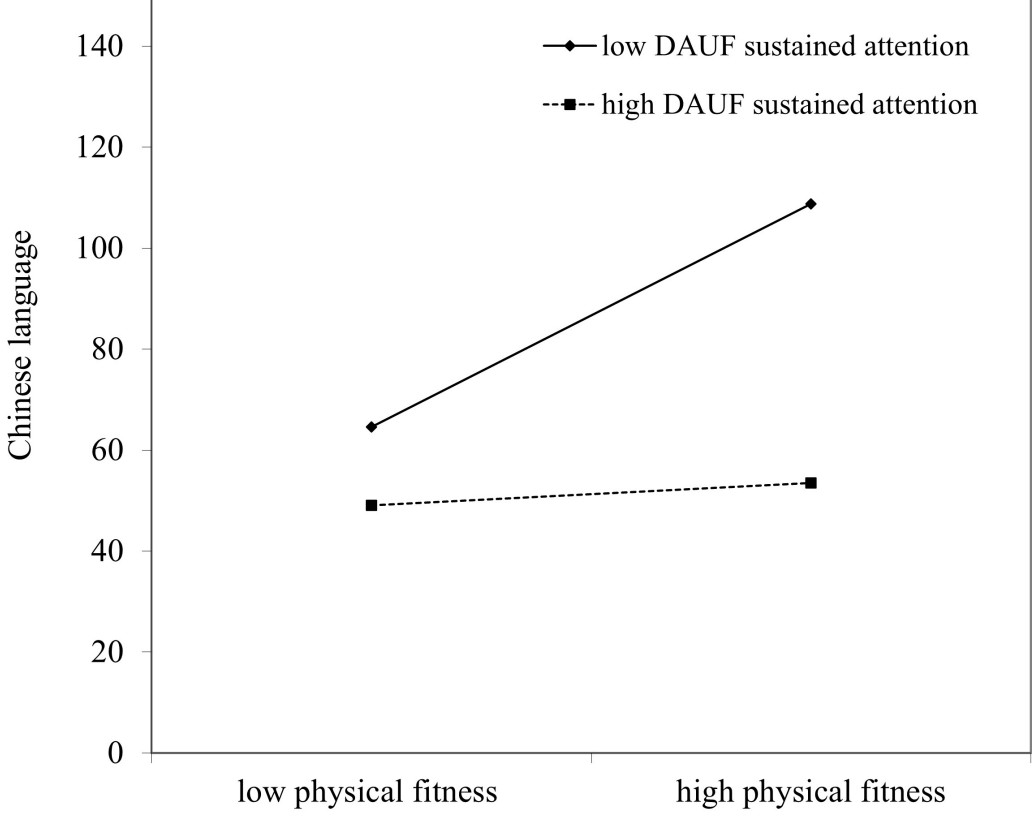

**Figure 1  The moderating effects of DAUF sustained attention on physical fitness-Chinese language.**

a moderating effect. Children with LD who have better physical fitness tend to achieve higher mathematics scores if their DAUF sustained attention levels are low.

## DISCUSSION

This study aimed to examine the modulatory effects of DAUF sustained attention on academic performance in children with LDs. Unlike previous experimental studies that

**Table 4  Summary results of the moderating effects for mathematics.**

| Variables | Step 1: control variable | | | Step 2: direct effect | | | | Step 3: interaction effect | | | |
|---|---|---|---|---|---|---|---|---|---|---|---|
| | $\beta$ | $t$-value | $R^2$ | $\beta$ | $t$-value | $R^2$ | $\Delta R^2$ | $\beta$ | $t$-value | $R^2$ | $\Delta R^2$ |
| Age | −0.02 | −0.14 | | −0.04 | −0.40 | | | −0.04 | −0.36 | | |
| BMI | 0.11 | 0.94 | | 0.13 | 1.18 | | | 0.17 | 1.68 | | |
| Physical activity | 0.12 | 1.02 | 0.033 | −0.02 | −0.19 | 0.256 | 0.223** | −0.05 | −0.48 | 0.356 | 0.100** |
| Physical fitness | | | | 0.12 | 2.16* | | | 2.67 | 3.42** | | |
| DAUF sustained attention | | | | −0.43 | −3.87** | | | 3.97 | 2.83** | | |
| Interaction | | | | | | | | −5.00 | −3.15** | | |

Notes.
*$p < 0.05$.
**$p < 0.01$.

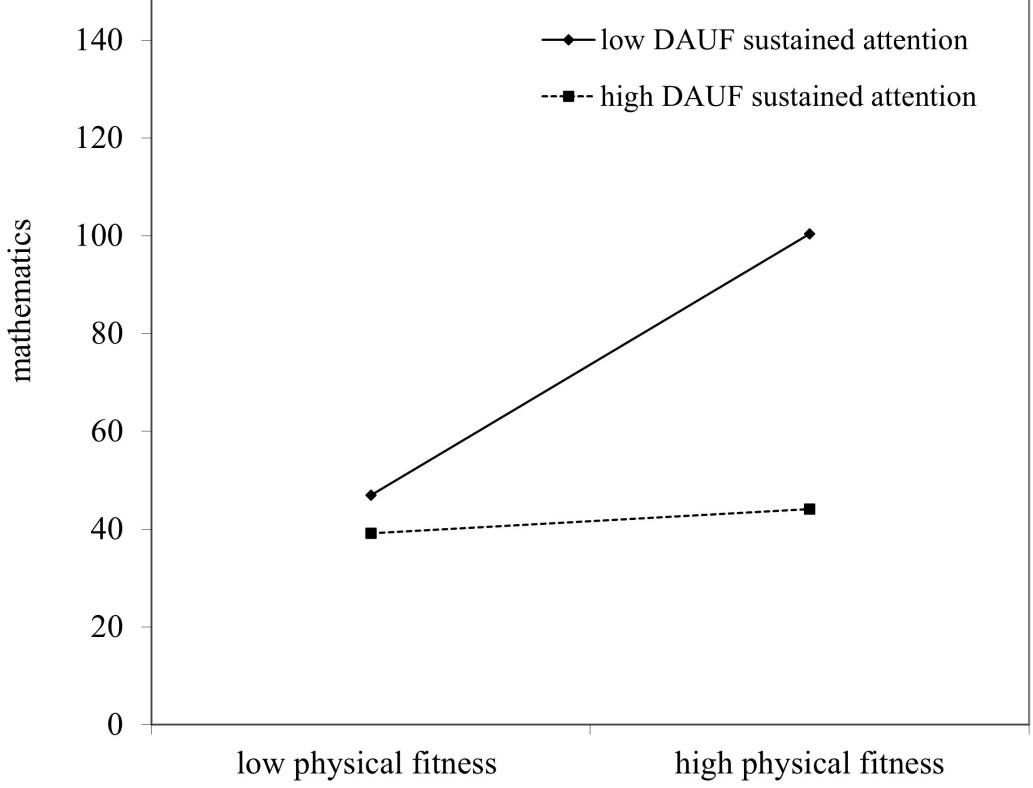

**Figure 2  The moderating effects of DAUF sustained attention on physical fitness-mathematics.**

primarily established causal relationships, our cross-sectional approach reveals predictive patterns among these variables, offering a complementary perspective on how these factors interact in naturalistic settings. Our findings reveal a complex interplay between physical fitness, sustained attention, and academic achievement in this population, offering new insights into the cognitive processes underlying learning in children with LD.

## The complex interplay of physical fitness, DAUF sustained attention, and academic performance

Our correlation analysis demonstrated significant positive relationships between Chinese language and mathematics performance and various components of physical fitness (cardiovascular fitness, muscular strength, and muscular endurance) in children with LD. The significant negative correlations between DAUF sustained attention and academic performance ($r = -0.51$ & $-0.43$, $p < 0.01$) indicate that students with better sustained attention (lower DAUF scores) demonstrated better academic achievement in both Chinese language and mathematics. Conversely, we observed a negative correlation between academic performance and DAUF sustained attention. This negative correlation reflects that better sustained attention (lower DAUF scores, as DAUF measures reaction time) was associated with better academic performance, which aligns with *Huang et al. (2020)* finding that sustained attention is a key predictor of academic success.

These results align with previous studies on the relationships between sustained attention, language skills, and mathematical abilities (*Huang et al., 2020*), while also extending our understanding of how physical fitness contributes to cognitive function in this population.

A key finding of our study is that DAUF sustained attention moderates the relationship between physical fitness and academic performance, particularly in Chinese language and mathematics. Interestingly, children with higher physical fitness levels tend to perform better academically when their sustained attention levels are lower. This finding may indicate a compensatory mechanism. Physical fitness might mitigate attention deficits through enhanced executive functioning. Alternative explanations include differences in cognitive load management and enhanced executive functions that help compensate for attention deficits. We acknowledge that other variables not measured in this study, such as motivation and self-regulation, could also influence this relationship. These findings align with previous research demonstrating the positive effects of physical fitness on cognitive function and academic performance in children with neurodevelopmental disorders (*Álvarez-Bueno et al., 2017*).

The association between physical fitness and cognitive function exhibited domain-specific relationships, with aerobic fitness influencing sustained attention tasks. These differential effects underscore the discrete impacts of different fitness components on cognitive outcomes. These findings imply a need for an individualized approach when designing interventions for children with LD. Thus, the present findings align with previous research on the positive relationship between physical fitness, attention, and academic performance in children with LD (*Pontifex et al., 2013*; *Fedewa & Ahn, 2011*). Notably, our study does highlight the specific moderating role of sustained attention, suggesting that children with lower levels of DAUF sustained attention may benefit more from fitness interventions. A complex interaction of cognitive compensation mechanisms was shown, where physical fitness may mitigate attention deficits by enhancing other cognitive functions (*Pontifex et al., 2019*; *Stern, 2009*). Brain plasticity and neurogenesis were enhanced through physical activity, particularly in learning-memory regions (*Mandolesi et al., 2018*). This mechanism facilitates cognitive flexibility and academic performance in children with LD.

 

Our present findings emphasize further exploration of these complex relationships among cognitive capacities, physical fitness, and academic performance in children with LD. Such understanding facilitates more effective intervention protocols using physical activity for cognitive-academic growth.

### Extending previous research: new insights into LD and cognitive functioning

Our findings highlight the complex interaction between physical fitness, DAUF sustained attention, and academic performance in children with LD. The moderating role of DAUF sustained attention appears more pronounced among students with lower levels of sustained attention derive greater benefits from physical fitness interventions. This supports the need for more tailored pedagogical approaches, aligning with recent research on cognitive development in this vulnerable population (*Peng & Kievit, 2020*; *Edwards et al., 2021*).

These empirical findings have pedagogical implications. Children with lower levels of DAUF sustained attention may benefit most from combining physical activity with attention-oriented cognitive exercise. For example, implementation of structured physical activities, such as team sports or aerobic exercises, paired with attention-oriented cognitive tasks, can improve both physiological and cognitive outcomes. As evidenced by meta-analyses (*Fedewa & Ahn, 2011*; *de Greeff et al., 2018*), such interventions yield the positive impact of physical activity on cognitive outcomes and academic performance, particularly in preadolescent populations.

Our research proposes a tailored intervention strategy based on individual levels of DAUF sustained attention to optimize academic performance in children with LD. This multifaceted approach, which integrates physical activity with attention-oriented cognitive tasks, aligns with the need for individualized interventions in special education (*Horowitz, Rawe & Whittaker, 2017*). Research has shown that such strategies can facilitate both executive functions and attention (*de Greeff et al., 2018*), ultimately leading to better cognitive and academic proficiency.

### Limitations and Recommendations

While this study provides important insights, certain methodological constraints warrant examination. First, while the cross-sectional design enables us to examine predictive relationships among multiple variables simultaneously—something experimental designs often cannot accomplish—it restricts our ability to infer causality between physical fitness, DAUF sustained attention, and academic performance. Future longitudinal studies will be crucial for determining the directionality of these relationships and how they evolve over time. Second, the relatively small sample size, drawn from children with LD within northern Taiwan, may restrict the generalizability of the findings to other populations, varied cultural contexts, or even other heterogeneous LD subgroups. Educational systems, diagnostic practices, and cultural attitudes toward physical activity and academic achievement differ significantly across countries, potentially affecting the relationships observed in this study. Future studies should include more diverse samples across different educational contexts and geographical areas to ensure broader applicability.

Third, the study did not account for confounding variables such as socioeconomic indicators, domestic environment, or accessibility of physical education resources, all of which could have an influence on both physical fitness and academic outcomes. Finally, our evaluation methodology of this present study focused primarily on academic performance in Chinese language and mathematics, which may inadequately capture the diverse spectrum of cognitive and academic outcomes relevant to children with LD. Future studies should incorporate a broader set of academic measures, including science, social studies, and other subjects, to better understand the comprehensive effects of physical fitness across different academic domains, which may exhibit different patterns of association.

**Future Research Directions**

Several directions for future research are identified as follows. First, causal relationships require methodical manipulation of fitness parameters to demonstrate the direct effects on DAUF sustained attention and academic achievement. Second, exploring neurobiological adaptations such as hippocampal plasticity in our targeted population may reveal underlying mechanisms through physical activity enhancing cognitive functions. Therefore, neuroimaging techniques like functional MRI or EEG could be employed to track how physical activity influences brain regions involved in attention, memory, and executive functioning.

Third, extending this research to children with other neurodevelopmental disorders, such as ADHD or autism spectrum disorder (ASD), could broaden our understanding of the role of physical fitness in cognitive and academic functioning across clinical populations, potentially revealing shared or unique mechanisms. Fourth, this study only explores the relationships between LD and the variables, without addressing causal relationships within different LD subtypes. Since LD often overlaps with various neurodevelopmental disorders, different outcomes may emerge across subgroups. Therefore, future research is recommended to conduct comparative studies on different LD subtypes to enhance the comprehensiveness of this research context. Finally, a more robust research design should be adopted to assess cognitive and academic parameters. Time-series studies could uncover the causality and temporal dynamics between physical fitness, sustained attention, and academic performance in such population. Additionally, the heterogeneous nature of LD across different LD subtypes (*e.g.*, dyslexia, dysgraphia, dyscalculia) warrant further investigation. These multifaceted research designs will provide a more nuanced understanding of the complex dynamics between physical and cognitive factors in these distinct populations.

## CONCLUSION

This cross-sectional study investigated the moderating role of sustained attention in the relationship between physical fitness and academic performance in children with learning disabilities. Our key findings demonstrate that sustained attention significantly moderates the association between physical fitness and academic achievement in both Chinese language and mathematics, with children having lower levels of sustained attention showing stronger benefits from higher physical fitness levels. These findings have important

educational implications, suggesting that physical fitness interventions may be particularly beneficial for children with LD who struggle with attention deficits, and highlighting the need for individualized approaches that consider cognitive profiles when designing physical education programs. Future research should employ longitudinal or experimental designs to establish causal relationships and explore how different types of physical activities might specifically enhance sustained attention in this unique population.

### Funding
The authors received no funding for this work.

### Competing Interests
The authors declare there are no competing interests.

### Author Contributions
- Hui-Ping Chang conceived and designed the experiments, performed the experiments, analyzed the data, prepared figures and/or tables, authored or reviewed drafts of the article, and approved the final draft.
- Tzu-Wen Lin conceived and designed the experiments, performed the experiments, analyzed the data, prepared figures and/or tables, authored or reviewed drafts of the article, and approved the final draft.
- Yi-Hsiang Chiu conceived and designed the experiments, performed the experiments, analyzed the data, prepared figures and/or tables, authored or reviewed drafts of the article, and approved the final draft.
- Chien-Chih Chou conceived and designed the experiments, performed the experiments, analyzed the data, prepared figures and/or tables, authored or reviewed drafts of the article, and approved the final draft.
- Jui-Fu Chen conceived and designed the experiments, performed the experiments, prepared figures and/or tables, and approved the final draft.
- Cheng-Chen Pan conceived and designed the experiments, performed the experiments, analyzed the data, prepared figures and/or tables, authored or reviewed drafts of the article, and approved the final draft.

### Human Ethics
The following information was supplied relating to ethical approvals (i.e., approving body and any reference numbers):

The Institutional Review Board (IRB-2017-044) at the University of Taipei.

### Data Availability
The raw measurements are available in the Supplementary File.

## Supplemental Information

Supplemental information for this article can be found online at http://dx.doi.org/10.7717/peerj.19549#supplemental-information.

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
