# Peer review of "Associations of physical fitness with sustained overt attention and academic performance in children with learning disabilities"

_PeerJ, doi:10.7717/peerj.19549_

## Round 0.1 · original submission · Major Revisions

All the reviewers have provided useful feedback. Please respond to them in your revisions

Reviewer 1 ·

Basic reporting

The authors should correct and improve the English. The theoretical background is sufficient, but the introduction regarding the connection between physical exercise and cognitive abilities should be enriched.

Experimental design

The method is well described, and the research question is well defined. The study has several limitations, but the authors have listed them at the end of the manuscript.

Validity of the findings

The aim of the paper is interesting; however, the methodology used is not sufficient to significantly advance scientific knowledge. Nevertheless, the paper contributes to the growing body of evidence highlighting the importance of considering physical exercise as a factor influencing neuropsychology and cognitive functioning.

Reviewer 2 ·

Basic reporting

- English language could be improved in a lot of different parts of the manuscript. Some sentence are left incomplete (i.e., line 62 "[...]a foundational cognitive resource (a that is missing here I think) serves as a core[...], line 105-107 reads incompletely, etc.")
- On line 70, the authors say that people with reading disabilities may show phonological processing and verbal working memory difficulties. But they also present attentional deficits, they could maybe add some literature about that. In general, the background provided seems a bit scarce (for example line 99 to 102 could be a bit more expanded).
- The structure of the manuscript is fine, even though the authors do not report the exact significance for the analysis in the text, just if it's p >.05, < .05 or < .01. In table 1, two different columns for SD and % could be used for clarity.

Experimental design

- While it's true that the research has a relevant meaning in the field, the methods and rationale behind could have been explained better. For example the authors investigate the physical fitness of the children, but they do not consider the social economic status of the families, indeed they cite them this themselves in the limit of the study.
- They also never mentioned if there were inclusion or exclusion criteria for the children. Since they underwent a physical fitness battery, I would assume that's why they did the medical history questionnaire.
- It's not really clear if the physical and cognitive assessments were done in the same morning, if so were possible fatigue effects kept into account?

Validity of the findings

- The conclusions could be made more clear and linked to a better explained introduction.
Some rationale is missing especially when explaining the "knowledge gap" (line 105). Some more examples could be given in line 96, for example.

Reviewer 3 ·

Basic reporting

This is a very important and complex area of research and new information makes a valuable contribution.
1. In the measures section state how the DAUF is scored including clearly stating whether high scores represent higher sustained attention or lower sustained attention. Add a note under the tables stating this to help readers interpret the correlations. In the discussion make it clear what a negative correlation between the DAUF scores and the other variables means in terms of higher or lower sustained attention.
2. In the participants section give a list of the learning disabilities that the partciapnts had.
3. In the discussion and conclusion please make it clear that this cross-sectional study can only indicate the associations, and possibility of causal links, it is not able to test for causality.
4. Tables and graphs, Please correct DAUF in Figures 1, 2.
5. Page 22, The conclusion must be improved and limited to the results of this study. Please state the type of study undertaken, the key findings, then state the implications and finally give the sentence commenting on the future research.

Experimental design

The design is appropriate. Please note the advice to add to the measures section a statement clarifying what high scores on the DAUF represent so readers can easily interpret the negative correlations reported.

Validity of the findings

The findings are presented in detail.
Please consider the advice on the conclusion.
1. Page 22, The conclusion must be improved and limited to the results of this study. Please state the type of study undertaken, the key findings, then state the implications and finally give the sentence commenting on the future research.
2. Page 22, lines 378 – 380, Please clarify this sentence, possibly two separate sentences might be easier: “Thus, the theoretical framework of LD is enhanced by these findings, in which practical applications are also emerged through developing more effective and individualized educational strategies.”

Additional comments

Please review the grammar. Some examples are provided below, but the whole document must be reviewed. Some other comments are also offered below to enhance this manuscript.
Abstract
1. page 6, line 28, the word sustain should be sustained: “…sustained attention recognized as a key predictor…” The same error is repeated on lines 32, 35, 41, 44.
2. Page 6, line 38, the word positive must be amended to positively: “as a dependent variable, significantly positively related to physical fitness…”
3. Page 6, line 39, the word negative must be amended to negatively: “…significantly negatively related to DAUF…”
4. Page 6, line 30, insert (LD) after the words learning disabilities: “…physical fitness and academic performance in children with learning disabilities (LD), and existing…”
5. Page 6, lines 43 – 45, The final sentence in the abstract doesn’t present the physical fitness as factor that this study appears to be promoting. Suggested change: “These findings have major implications for child development, emphasizing the key role of physical fitness in the beneficial effects of sustained attention on academic achievement.”
Introduction
6. Page 8 line 49, Please correct the sentence by deleting the word characterized: “Learning disabilities (LD) are neurobiological disorders that make learning …”
7. Page 8, lines 61 – 63, Please insert the word that and delete either efficient OR effectiveness: “The higher-order cognitive mechanisms rely intrinsically on sustained attention, a foundational cognitive resource serves as a core cognitive element for prolonged mental focus, to facilitate efficient processing effectiveness”, for example: “The higher-order cognitive mechanisms rely intrinsically on sustained attention, a foundational cognitive resource that serves as a core cognitive element for prolonged mental focus, to facilitate efficient processing.”
8. Page 8, line 66, Please rephrase this sentence: “LD demonstrates substantial cognitive diversity across its spectrum.” For example: “There is substantial cognitive diversity across the spectrum of LD.”
9. Page 9, lines 85, 86, insert a supportive reference for the sentence: “The relationship between ……LD population (REF).
10. Page 9, lines 91, 92, Please rephrase this sentence, it is unclear. “This knowledge gap needs to be addressed to illuminate cognitive profiles and attention related deficits in children with LD.”
11. Page 10, lines 105 – 107, Rephrase this sentence for clarity: “To address these knowledge gaps, how sustained attention might play a role of moderating relationship between physical fitness and academic achievement in two fundamental academic domains (i.e., Chinese language and mathematics) were addressed.”
Discussion
12. Page 18, lines 281 – 283, This sentence: “Conversely, we observed a negative correlation between academic performance and DAUF sustained attention.” appears to contradict the sentence on Huang et al, 2020, “Physical fitness has consistently been linked to cognitive and academic performance, with sustain attention recognized as a key predictor of academic success (Huang et al., 2020).” “Conversely, we observed a negative correlation between academic performance and DAUF sustained attention.” Please clarify this.

·

Basic reporting

Strengths:
- The manuscript is generally well-structured and follows PeerJ's formatting requirements.
- The introduction provides a good background on learning disabilities (LD) and their relationship with physical fitness and cognitive functions.
- Figures and tables are relevant, labeled appropriately, and contribute to the clarity of the findings.
- The study is well-referenced, drawing from relevant literature on physical fitness, attention, and academic performance.

Areas for Improvement:
Clarity and Grammar:
- The manuscript contains multiple grammatical errors and awkward phrasing, which impact readability. For example:
"Sustain attention" should be "sustained attention" throughout the manuscript.
"This study enrolled 72 elementary school children with learning disabilities (34 girls, mean age = 11.03 years ± 0.82), who participated the assessment of physical fitness" should be revised to "This study enrolled 72 elementary school children with learning disabilities (34 girls, mean age = 11.03 years ± 0.82), who participated in the assessment of physical fitness."
- The phrase "Chinese language and mathematics tests were administered to assess academic performance" would be clearer as "Academic performance was assessed using Chinese language and mathematics tests."

Literature Review:
- While the introduction covers important prior research, there is little discussion of potential contradictory findings in the literature. Including studies that present differing views would help balance the discussion.

Figure and Table Presentation:
- Figures and tables are well-structured but should include more explanatory captions to make them self-explanatory. For example, Figure 1 and Figure 2 would benefit from a brief description of the moderating effect observed.

Experimental design

Strengths:
- The research question is clearly defined and contributes to an identified knowledge gap concerning the relationship between physical fitness, sustained attention, and academic performance in children with LD.
- The study includes a reasonable sample size (72 children), though slightly small, it is justified given the targeted population.
- Ethical approval and informed consent procedures are appropriately mentioned.

Areas for Improvement:
Methodological Rigor:
- The selection criteria for participants should be more explicitly stated. While the study specifies exclusion criteria (e.g., autism spectrum disorder and ADHD diagnoses), it is unclear whether children with other neurodevelopmental conditions (e.g., dyslexia, dyscalculia) were included.
- The DAUF Sustained Attention Test is a validated measure, but more information on its reliability and validity in LD populations would strengthen the methodology.
- The study should clarify whether the academic performance assessments were standardized across all participating schools.

Physical Fitness Measurements:
- The authors used cardiovascular fitness, muscular power, muscular endurance, and flexibility as indicators of overall physical fitness. However, it is unclear why agility and coordination were not considered, as these may be particularly relevant in children with LD.
- More details on the standardization process for fitness tests should be provided. Were the tests conducted by trained assessors? Was intra-rater or inter-rater reliability assessed?

Validity of the findings

Strengths:
- The findings contribute to the literature on physical fitness and cognitive function in children with LD.
- The results are statistically significant and aligned with previous research in related fields.
- The discussion is insightful and proposes practical implications for educational and intervention strategies.

Areas for Improvement:

Interpretation of Results:
- The authors conclude that physical fitness has a compensatory effect on academic performance in children with low sustained attention. However, this claim should be made more cautiously. Other potential explanatory variables (e.g., motivation, self-regulation) should be acknowledged.
- The claim that "children with higher physical fitness levels tend to perform better academically when their sustained attention levels are lower" is counterintuitive and needs more theoretical grounding. Alternative explanations (e.g., differences in cognitive load, the role of executive function) should be explored.

Causal Inference:
- The cross-sectional design does not allow for causal conclusions. While the authors acknowledge this limitation, they should discuss potential methods for future longitudinal studies to establish causality.
The findings should be framed within the context of correlation rather than causation.

Generalizability:
- The sample consists of children from a specific educational system in Taiwan. The authors should discuss whether findings can be applied to other cultural or educational contexts.
- The study focuses only on two academic subjects (Chinese language and mathematics). Other subjects (e.g., science, social studies) may exhibit different patterns of association with physical fitness.

---

## Round 0.2 · Minor Revisions

The authors have addressed most of the reviewers' comments. Some minor aspects should be considered before accepting the paper for publication.

Reviewer 2 ·

Basic reporting

1. "We appreciate the reviewer's careful attention to the English language throughout our manuscript. We have thoroughly revised the paper to improve clarity and grammatical accuracy. Specifically: 1. We have corrected the incomplete sentence on page 3, line 64 by adding the missing relative pronoun "that". 2. We have revised the incomplete sentence structure on page 5~6, lines 123-125 to improve clarity. 3. We have conducted a thorough review of the entire manuscript to identify and correct any other grammatical issues, awkward phrasing, or incomplete sentences to ensure the paper meets high standards of academic English."

I appreciate the authors’ efforts to improve the English language throughout the manuscript. However, a few minor issues still need to be addressed, for example on line 76-78 it reads "In addition, research reveals that extensive neurobiological patterns between LD and various developmental disorders, especially in ADHD, and that 30-50% of children with ADHD meeting LD criteria (DuPaul et al., 2013)." the sentence seems incomplete. Please re-read the whole manuscript one last time.

2. "We thank the reviewer for highlighting these important areas for improvement. We have expanded this section to include literature on attention deficits in reading disabilities (See Page 4 line75-76, adding Provazza et al., 2019) and to provide a more comprehensive background on the research gaps (See Page 5 line106-117, adding Huang et al., 2020; Padmakumari & Raj, 2016; Raine et al., 2013). "

The response and corresponding revision are appropriate and adequately address the original concern.

3. "We appreciate the reviewer's suggestion regarding Table 1. After further consideration, we decided to maintain both SD and % columns since we have now included learning disability types (Language and Mathematics) as categorical variables alongside gender. This format allows for clear presentation of both continuous variables (with SD) and categorical variables (with %) in a consistent structure. We have improved the table's layout to enhance clarity, including a note specifying that "learning disability in language including dyslexia and dysgraphia."

I appreciate the authors’ response and the improvements made to the table layout. However, to clarify my original comment: my suggestion was that the current format, which combines SD and % into a single column, may be somewhat unclear for readers. Separating SD and % into two distinct columns could improve clarity, particularly when presenting both continuous and categorical variables.

That said, I respect the authors’ decision to maintain the current structure.

Experimental design

1. We appreciate the reviewer's concern about socioeconomic status (SES). We have addressed this by:

1. Adding information in the Methods section (page 6 lines 131-132 ) clarifying that participants were recruited from schools within the same district in Taipei City, representing a relatively homogeneous urban environment.

2. We have already acknowledged this limitation in our original manuscript (p. 17, lines 406-408): "Third, the study did not account for confounding variables such as socioeconomic indicators, domestic environment, or accessibility of physical education resources, all of which could have influence on both physical fitness and academic outcomes."

The response and corresponding revision are appropriate and adequately address the original concern.

2. We appreciate the reviewer highlighting this oversight. Our manuscript did include exclusion criteria (p. 10, lines 233-237): "Two children with autism spectrum disorder and five children attention deficit hyperactivity disorder were excluded subsequently from the study due to documented diagnoses." However, we agree this could be more clearly presented. We have repositioned this information to clearly identify it as formal inclusion/exclusion criteria and clarified that the medical history questionnaire was used as part of our screening process to ensure participant eligibility and safety for the physical fitness assessments.

The response and corresponding revision are appropriate and adequately address the original concern.

3. Thank you for this important methodological question. We can confirm that the cognitive assessment (DAUF Sustained Attention Test) and physical fitness assessments were conducted on different mornings, not on the same day. We have clarified this in the Experimental Procedure section (p. 10, lines 223-227) to state: "On the following day after anthropometric measurement, each participant completed the DAUF Sustained Attention Test. Physical fitness assessments were conducted on a separate morning, with participants instructed to refrain from strenuous physical activity on the day of assessment." This scheduling approach eliminated concerns about potential fatigue effects between cognitive and physical assessments.

The response and corresponding revision are appropriate and adequately address the original concern.

Validity of the findings

No comment.

Reviewer 3 ·

Basic reporting

The authors have addressed each point I raised in the review. I am satisfied with their new version of the manuscript.

Experimental design

The authors have addressed each point I raised in the review. I am satisfied with their new version of the manuscript.

Validity of the findings

The authors have addressed each point I raised in the review. I am satisfied with their new version of the manuscript.

·

Basic reporting

Authors have adequately addressed all the comments in the revised version of the manuscript. Therefore, I have no further comments.

Experimental design

Authors have adequately addressed all the comments in the revised version of the manuscript. Therefore, I have no further comments.

Validity of the findings

Authors have adequately addressed all the comments in the revised version of the manuscript. Therefore, I have no further comments.

Additional comments

Authors have adequately addressed all the comments in the revised version of the manuscript. Therefore, I have no further comments.

---

## Round 0.3 · accepted · Accept

The author have addressed the requested points and the paper can be accepted in the present form.

Reviewer 2 ·

Basic reporting

The response and corresponding revisions are appropriate and adequately address the original concern regarding clarity and grammar accuracy.

Experimental design

No comment.

Validity of the findings

No comment.

Additional comments

No comment.